# Applying a User Centred Design Approach to Optimise a Workplace Initiative for Wide-Scale Implementation

**DOI:** 10.3390/ijerph19138096

**Published:** 2022-07-01

**Authors:** Ana D. Goode, Matthew Frith, Sarah A. Hyne, Jennifer Burzic, Genevieve N. Healy

**Affiliations:** 1School of Human Movement and Nutrition Sciences, The University of Queensland, Brisbane, QLD 4072, Australia; g.healy@uq.edu.au; 2Kin8, The Commons QV, 3 Albert Coates Lane, Melbourne, VIC 3000, Australia; mfrith@kin8.com.au; 3SSH Design, Fitzroy North, Melbourne, VIC 3068, Australia; sarah.hyne@gmail.com; 4School of Public Health, The University of Queensland, Brisbane, QLD 4072, Australia; jenburzic@gmail.com

**Keywords:** user centred design, RE-AIM, implementation, workplace, sedentary, health promotion, behaviour change, champion, peer-led

## Abstract

Translation of an effective research intervention into a program able to be implemented in practice typically requires adaptations to ensure the outcomes can be achieved within the applied setting. User centred design (UCD) methodologies can support these iterative adaptations, with this approach being particularly well suited to peer-led interventions, due to a focus on usability. We describe and reflect on the UCD approach that was applied to optimise an online, peer-led workplace health promotion initiative (BeUpstanding: ACTRN12617000682347) to be suitable for wide-scale implementation and evaluation. Optimisation was aligned against the indicators of the RE-AIM (reach, effectiveness, adoption, implementation, maintenance) framework, with UCD methodologies (discovery interviews, persona and scenario mapping, facilitated workshops, surveys and prototyping) employed to enhance the program according to all RE-AIM dimensions. The core team (content experts, implementation scientist, interaction designer, software developer, business developer) worked closely with policy and practice partners and end users (workplace champions, management and staff) to iteratively develop and test across the RE-AIM indicators. This description and reflection of the process of applying UCD and the RE-AIM framework to the optimisation of BeUpstanding is intended to provide guidance for other behaviour change research adaptations into practice.

## 1. Introduction

Adaptations of evidence-based health practices or interventions are almost always needed to maximise and maintain outcomes within applied settings [1,2,3]. Although many implementation and adaptation frameworks and models exist [4,5], to date there have been relatively few attempts to understand how and why adaptations occur, what informs them, and their ultimate impact. User-centred design may offer a methodology to support and understand rapid, continued and iterative adaptations to improve evidence-based health programs [6,7,8]. User centred design (UCD) is by its nature iterative and collaborative, bringing together research in human-computer interaction, user experience design, service design, behavioural science and cognitive psychology [9,10]. The main strength of this methodology is that it quickly identifies multiple stakeholders and/or end users and grounds the design of an innovation (such as an evidence-based health behaviour program) in information about them and their needs in context [11,12]. A collaborative, participatory and partnered approach of co-creation is used to consider and incorporate these multiple perspectives [13]. This early identification and engagement process increases the chance that a program will be taken up and used by the people it was designed for. This approach is likely to be particularly well suited to interventions that are peer-led, where feasibility and usability are of prime importance for intervention delivery and evaluation.

A user-centred design approach was used for the optimisation of the evidence-informed BeUpstanding program and accompanying BeUpstanding Champion Toolkit to ensure it was fit-for-purpose for a national implementation trial. The BeUpstanding program is a workplace health initiative designed to support work teams to sit less and move more for their health and wellbeing [14]. The program employs a ‘train the champion’ approach, with workplace champions accessing an online toolkit with a step-by-step guide and accompanying resources (e.g., videos, email templates, posters, survey links) to support the set up, delivery and evaluation of the program within their work team. The goals of the program are to raise awareness of the benefits of sitting less and to create a supportive workplace culture where sitting less and moving more is the norm.

From its inception, the BeUpstanding program was designed with wide-scale uptake and implementation in mind; employing an iterative, phased process of development and adaptation [15,16,17]. These planned phases and corresponding development and testing have been grounded in the well-established RE-AIM (Reach, Effectiveness, Adoption, Implementation, Maintenance) framework [18]. Due to its ease of use and balanced focus on internal and external validity, RE-AIM has been used extensively to guide the design and evaluation of programs in applied contexts [19]. It can also be used to highlight common elements that will require attention from researchers and program designers to promote the success of programs in these contexts [20]. Details of the phased approach of adapting BeUpstanding according to the RE-AIM framework have been reported elsewhere [15,16], with findings summarised below.

Phase 1 focussed on the development of the ‘BeUpstanding Champion Toolkit’ and establishing research-government/industry partnerships [15]. During this phase, intervention material and protocols from the researcher-delivered *Stand Up Australia* program of workplace intervention research [21,22] were adapted to be more suitable for wide-scale delivery, with a key adaptation being the change in delivery agent from trained researcher to workplace champion. Use of such peer champions has been shown to be an effective way to disseminate knowledge within workplaces, with champions acting as necessary role models and drivers for staff participation and team change [23,24,25]. This “train-the-champion” model also aligned with the preferences of the workplace policy/practice partner that provided seed funding to enable this adaptation process.

Phase 2 tested the feasibility and acceptability of this champion-led approach via the evaluation of a beta (test) version of the program. Findings showed that this approach was acceptable and feasible for champions [26], with findings also showing the program was effective in reducing self-reported workplace sitting time [17]. This phase specifically tested the feasibility of program delivery by a workplace champion. It did not design or test features that would make it easily adopted or evaluated at scale (i.e., researchers recruited champions and guided evaluation).

Phase 3 involved the development and testing of features and functions to enable workplaces and champions to find out about the program, as well as evaluate it without direct researcher involvement [16]. These features and functions included the integration of online on-boarding pages and evaluation components (e.g., surveys) via a bespoke technology platform, with the intention that program evaluation became a ‘core component’ for champions to lead. Correspondingly, changes to the user interface were made to highlight the core intervention elements, and incentives were added to the user experience for completing these tasks, including a customisable poster template and real-time data reporting of the main behavioural outcome (workplace sitting, standing and moving). The updated website and toolkit was made freely accessible across Australia with limited and targeted promotion by established partners via a “soft-launch” on 1st September 2017. In user-centred design terms, this could be considered the minimal viable product (MVP), where the purpose was to test the new on-boarding and recruitment channels and the new integrated delivery and evaluation platform with end-users “in-the-wild”. The ultimate goal of this phase was to inform further redesign to optimise the program prior to it’s’ evaluation in the context of a planned national implementation trial (Phase 4) [14].

The aim of this paper is to describe the redesign process that was conducted from Phase 3 to Phase 4 in order to make the program suitable for wide-scale implementation by workplace champions according to the indicators of RE-AIM. Objectives are to: (1) provide examples of the application of UCD strategies for the redesign of the program and (2) highlight priorities for redesign according to stakeholders and (3) reflect on the process of optimization including the key elements required for success. Provision of this detailed description of the optimisation process, and the UCD methodology underpinning the process, is intended to provide guidance for bringing other adaptations of behaviour changing programs into practice.

## 2. Methods

### 2.1. Methodological Approach and Participants

A UCD methodology [27] was applied to optimise the BeUpstanding Champion Toolkit for wide scale implementation and evaluation. The research team contracted experts in design thinking and interaction design (SH), and business and product development (MF), to form the core expert team which included an implementation scientist (AG). The core expert team led the iterative process of redesigning the toolkit. The primary stakeholders involved in the redesign were an expert interdisciplinary team that included foundational members involved in the research to practice translation process. These foundational stakeholders included a software developer, and state, national policy and practice partners. Workplace end users including champions, management and staff were also represented (Table 1). The toolkit’s online registration form provided details for ‘user’ organisations to be recruited via email, with ‘non user’ organisations recruited through partner channels via email. Ethics approval was granted by the Human Research Ethics Committee of The University of Queensland (#2016001743).

### 2.2. Redesign Process and Data Collection

The study process was underpinned by three key design phases: ‘inspiration, ideation, implementation’, with all phases overlapping and being iterative in nature [28]. The *inspiration phase* [10] focused on identifying the problems inhibiting the BeUpstanding program and toolkit from being fit-for-purpose for the national implementation trial. These factors were identified according to the RE-AIM framework, as described previously [16]. Multiple data sources were used, including the online champion and staff program survey and website analytic data, as well as direct discovery interviews with end users (described further below). During the *ideation phase*, the core team then synthesised and interpreted this feedback in relation to the design elements. User journey maps and personas of workplaces, champions, staff and researchers were developed to draw insights and identify needs, challenges and any required actions. Mock-ups of the user journeys (champion, manager, staff) for the implementation trial were developed, as were wireframes and prototypes of the new elements for the toolkit including mixed-media promotional and program materials and data insights and reports. In line with recommendations [29,30,31], the core team took into account multiple user perspectives including staff, team champions and management, partners, researchers and resource constraints. The prototypes and redesigned assets were then presented and workshopped with the core evaluation group and partners as part of the *implementation phase*, with workplace users provided with the opportunity to provide written and verbal feedback on prototypes following the adaptions made from the workshops and meetings. The final redesigned elements were then integrated into the toolkit by the software developer for use in the national implementation trial. Workplace users were informed of the updates to the toolkit and program via the existing communication pathways (program newsletter, blog article on the dedicated program blogsite, and promotion through partner channels such as their websites and e-mailing lists).

The process of gaining feedback was iterative, and included multiple stakeholders as outlined in Table 1. Numerous methods [10] were used to generate, collect, and synthesise ideas and feedback provided from the multiple stakeholders (workplace users; practice and policy partners; core expert team) across these three phases. These are described in detail below:

#### 2.2.1. Workplace Users

*Online surveys*: Open-ended responses in the online surveys (champions; staff) were collected through the BeUpstanding toolkit and supporting implementation platform on what worked and didn’t work/could be improved. These were reviewed and synthesised.

*Analytics*: Objective user data (e.g., tasks completed by user champions) was collected automatically by the implementation platform, while information such as page visits and recruitment channels were evaluated through Google Analytics. This information was interpreted by the core team to understand challenges, and what was and what was not working as intended.

*Adhoc email and phone communication*: champions who acted in the champion and champion/overseer roles were purposely sampled from a mix of organisations (i.e., private sector, government, small and large organisational size) and across Australian states. They were asked to provide iterative feedback on the program and various elements that were redesigned (e.g., prototypes of data reports).

*Direct discovery interviews and field visits*: three organisations were chosen to conduct discovery research across management, champion and staff users. Workplaces were deliberately recruited to ensure a representation of engagement with the program. That is, one workplace was selected that was actively using the program and the others had not implemented the program, despite signing up. The interviews were conducted in-person by the core team (a project manager and honours level student). They followed a semi-structured interview script developed for the purposes of this work by the expert team (i.e., implementation scientist, content expert, product developer). Interviews were recorded using mobile application software, and field visit notes were collected and compared across three visiting team members. The interviews aimed to understand how the user was engaging, or how they would engage with the program. For existing champions, the questions centred on understanding how they were using the program and any changes they were making (e.g., to the delivery; to the materials) to enable suitability for their team. For management, the questions focused on understanding the uptake of the program and the information they needed to inform return-on-investment. For staff, the questions centred on their experience with BeUpstanding. For non-users, the questions were designed to understand what would help them engage with and deliver BeUpstanding in the future and any current barriers to implementation.

#### 2.2.2. Practice and Policy Partners

*Stakeholder meetings*: Formal six-weekly meetings (via video conferencing or phone) with the core research team and representatives from the policy and practice partner organisations were held, with a standard agenda item covering updates on the redesign. Where appropriate and needed, prototypes were shown via screen sharing technology or email, with feedback sought and encouraged. All stakeholder meetings were recorded and minuted.

*Adhoc email and phone communication*: Depending on the focus, at least one team member from each of the policy and practice partners was asked to provide feedback on redesigned program elements. For example, for the promotional materials and collateral that were designed to enhance reach and adoption, feedback from their communications and marketing team was provided. This happened on a needs basis.

#### 2.2.3. Core Expert Team

*Researcher meetings*: The research team held weekly project meetings, with a standing agenda item to review feedback collected via website analytics, adhoc feedback, survey data and interviews, as well as redesigned toolkit elements. Researchers led the collection of data in the inspiration phase.

*Intensive product development workshop*: A 1.5 day intensive workshop with the business and product developer and implementation scientist was conducted during the ideation phase. Personas of workplace champions, staff and researchers were developed to draw insights and identify needs, concerns and challenges.

*Intensive design workshops*: The interaction designer and implementation scientist held two, 1 day workshops to discuss, develop and prioritise ideas to redesign the toolkit in preparation for the implementation trial phase. Personas were further developed and journey maps and low fidelity prototypes were ideated.

*Adhoc email and collaboration tools*: The core expert team met regularly on a needs basis and used online team collaboration software (e.g., Slack, Slack Production 4.17.1 64-bit, Slack Technologies, San Francisco, CA, USA, InVision, Invision V6, InVision Enterprise, New York, NY, USA, Trello, Trello 4.3, Fog Creek Software, NewYork, NY, USA) to review designs and provide feedback. Ongoing iterative ideation and development also occurred informally via email and team meetings

*High fidelity prototype testing*: Testing was conducted by the core team once the software developer had integrated changes into the toolkit. Further redesign was communicated via email and in-person during meetings with the implementation scientist, interaction designer and software developer.

### 2.3. Analysis

Data are reported descriptively, in line with key frameworks as appropriate. Qualitative data were reviewed and synthesised thematically, with quotes used to highlight key user opinions as appropriate.

## 3. Results

A total of 113 Australian organisations (135 toolkit users) took part in the soft launch or ‘early adopters’ phase, providing at least sign-up data [16]. Workplace audit data was obtained from 71 champions, while staff survey data was obtained from 337 staff at pre-program and 167 staff post-program. Program outcomes according to RE-AIM are reported elsewhere [16]. Seven purposely sampled workplaces (i.e., a customer panel comprising 3 overseers and 12 champions from 5 Australian states) provided detailed adhoc feedback via email and/or phone. A further three purposely sampled workplaces (3 managers, 7 champions, 2 staff from Queensland: large government and nongovernment) and Victoria (small, non government) took part in the direct discovery interviews.

In line with our first objective, a summary of the UCD strategies employed during the design phases are presented in Table 2. The UCD strategies were used to identify key intervention features and processes that required redesign, according to the terms and definitions by Dopp et al., 2019 [12]. Overall, we used 22 of the 30 strategies, with the majority of the ones not used either not being applicable to this stage of optimisation or too time intense and costly to undertake. UCD strategies focussed on a broad array of characteristics, including the individual (e.g., personas), the intervention (e.g., co-creation sessions and prototyping), as well as the organisation (e.g., observational field visits).

Through the UCD strategies noted in Table 2, and synthesis of the feedback, key priorities for redesign emerged. In line with our second objective, a summary of these priorities according to key stakeholder or user are described in Table 3. In addition to the users highlighted in Table 3, commercialisation and business development partners also informed priorities for redesign, includingthe branding and logo, while the software developer advised on the redesign in relation to the constraints of the platform and the underlying data structure. The interaction designer advised on best practice in design thinking and user-centred design.

Table 4 summarises the design challenges and improvements made to optimise the program and the personnel support required to make the enhancements according to each of the five RE-AIM dimensions. Further detail is provided below.

### 3.1. Optimising Program Reach

Two major changes were undertaken to the toolkit and program design to enhance and understand program reach: professional design of the program materials, and modifications to the data collection platform.

*Professional design of materials*: Feedback from champions and staff revealed the importance of engaging, ‘fun’ and ‘fresh’ materials (e.g., emails, videos and posters) to convey the main messages of the program in order to encourage and motivate ongoing staff participation.

“...make it more fun and interactive”Champion 112

Input from policy and practice partners revealed some tension between end user’s desire for funny and irreverent content and their desire for an ‘official’ feel and focus on evidence-based content. Consequently, a professional videographer and graphic designer were employed to revise all online, printed and video resources to ensure consistent, engaging and evidence-informed resources. The research team devised the core messaging, with policy and practice partners signing off on end content.

*Modifications to the data collection platform*: The bespoke Wildfyre platform developed for the project integrated a survey management and data collection system which allowed the capture of team and staff characteristics to assess reach. Some champions struggled to initially identify the number of staff that would participate in the program as part of their team. Consequently, champions were provided with the ability to adjust their team numbers via their profile settings. Survey response rates, according to the team number, were displayed in the survey portal, providing another opportunity for the champion to review and check this important denominator.

### 3.2. Measuring Program Effectiveness

The majority of effectiveness indicators were adequately captured by the staff surveys at pre- and post-program [16]. However, the survey participation rate by staff was lower than desired (43%) in those teams taking part in the evaluation. When asked, champions reported hesitancy about sending the survey out and not wanting to “over-survey” staff. There were also concerns about data privacy and the length of the survey.

“People did not complete [the] survey because they feared what would happen to their data”Champion 121

To address this tension between the needs of the researchers to collect detailed and validated data, and the workplace concerns identified, education on the need and usefulness of surveys was incorporated, in line with the suggestion from champions. Champions were also provided with the opportunity to view the survey prior to sending the survey link to staff.

“Maybe a little summary at the beginning of the survey on what the survey was aiming to achieve and goals at the end of the survey”Champion 126

Managers and champions also reported wanting more access to engaging data insights to assist with further scale out across other work teams within the organisation and/or repeat rollout of the program. Incentives for data completion were added to help address this issue.

”…need for feedback on the activities that were put in place…we did not get outcomes”Champion 102

*Incentivising survey completion*: Incentives were added to the toolkit to encourage completion of key self-report surveys including more detailed real-time data insights at baseline and follow-up survey times displayed for the champion (e.g., the common barriers staff reported to sitting less and moving more were displayed in a table)/Tailored target response rates and feedback were also added (e.g., It’s great if at least 36/55 staff respond to this survey). A new, customisable poster and a 9-page downloadable bespoke data report available at program end, following completion of the core evaluation elements, were also developed and integrated into the toolkit. Decisions around the data to present in the bespoke report were informed by the feedback from champions and management. Here, the research team had used custom-generated Powerpoint slides to iteratively trial how to present the data collected in ways that were meaningful and appropriately interpreted by the end-users (champions; management; staff).

### 3.3. Enhancing and Tracking Adoption

To enhance and more readily track the adoption of the BeUpstanding program, three changes to the design of the toolkit were identified: clear presentation of the business case; clarification of the champion journey; and, streamlining onboarding. The need for targeted recruitment pathways was also acknowledged.

*Clear presentation of the business case*: Champion and management feedback highlighted the importance of presenting program outcomes that were of relevance to management when deciding to adopt the program in their organisation/team.

“need to be able to promote it more…to get management on board”Champion 102

Expected key program outcomes, such as behaviour change (i.e., change in workplace sitting time) were important, but so too were work outcomes including employee satisfaction, productivity and reduction in sick days. Consequently, the user’s onboarding journey included free downloads (e.g., “Dear Boss” letter) and an engaging short animation to help potential champions present the ‘business case’ to management to take part in the program and facilitate uptake.

*Champion journey*: Management and champion feedback indicated that they were keen to know exactly ‘what they were signing up for’ in delivering and evaluating the program. Management also wanted additional resources outside of the program information sheets to help recruit champions. A colourful two page and one page version of a ‘champion journey’ infographic (See Figure 1) was developed to outline the necessary time commitment, and key steps involved in this peer-led program.

*Streamlining onboarding*: To enable the automatic monitoring of the number and type of organisations coming into the program, an online sign-on form that captured basic team and organisational information was integrated during the soft launch phase. However, these forms were further streamlined and simplified to make it as easy as possible for a team representative (usually a champion) to complete the form before accessing the toolkit.

*Establishing targeted recruitment pathways*: For the program to be adopted at a national level and across all the sectors identified as priority targets, it was necessary to develop recruitment targets and referral pathways in collaboration with policy and practice partners. Planning sessions were held with each policy and practice partner to determine key audiences and potentially appropriate channels to promote the program. A tailored promotions and marketing plan with associated content and collateral was developed for each partner. Engagement with the partners was led by the implementation scientist (AG) and business strategist (MF).

### 3.4. Supporting Champion Implementation

Overall, champions found the toolkit and supporting materials easy to use and helpful to supporting implementation, but several elements were identified as not fit-for-purpose for either supporting or evaluating the program [16]. To address this, we developed new assets and made changes to the champion dashboard to help guide program delivery and enhance the data collection process.

*Development of additional assets and dashboard changes*: Champions often liked the supporting materials provided to help them implement the program, but commonly asked for additional collateral, particularly posters and tips sheets including ideas for strategies to keep messages ‘fresh’ and enhance staff engagement with the program.

“I have been using many of the posters available and think some more would be good, as I am rotating them around so they don’t become part of the furniture. I think it’s more useful for the posters to have a tip (e.g., walk over rather than sending an email) instead of a general statement (e.g., take a stand for your health)”Champion 168

In line with what was described for reach, new weekly posters and associated email content were developed and organised in a week-by-week guide to make it easier for champions to implement the main program messages. The “champion journey” asset (as mentioned above) was used to roadmap the main tasks involved in the implementation trial (see Figure 1). The numerous tasks involved in delivering and evaluating the program were grouped into key stages of the champion journey with simple terms such as ‘set up, stand up, wrap up’ ascribed to each stage to make it more memorable. The champion dashboard interface was changed to include visual signposts so champions could readily see what they had completed (e.g., flag and fade/ghosting) and identify ‘must do’ or core components (e.g., use of a star and bolded text). The flagging of core components was important to maintain fidelity but also to help champions tailor their BeUpstanding journey, as not all teams needed to complete all steps.

“We are a small dynamic team, in a small office and already had all the equipment necessary. We could have jumped a few of the steps in the program.”Champion 422

Design features were added to the well liked ‘push button’ design layout to encourage champions to complete tasks, check that they had completed them and stay on track (e.g., an autofill coloured program task bar). Visual cues (e.g., a preview of the bespoke report) were added to the dashboard to encourage champions to complete the workplace audit tool, which was a key planning step in the program.

*Enhancing implementation data collection and feedback*: A new champion-completion survey was added to the end of the program. This survey was designed to capture both champion and staff engagement and experiences with the program components and program costs. This data, coupled with other evaluation data, was used to generate a bespoke team performance report that the champion could then share with their team and management (see Effectiveness). To provide an incentive to complete the evaluation components, the report was only made available once there was sufficient evaluation data collected (i.e., a minimum of two staff had completed both the pre- and post-program staff surveys; the champion had completed the post-program survey).

### 3.5. Measuring Program Maintenance

The toolkit version used by early adopters (during the soft launch phase) did not include any measurement of program maintenance. Sustainability of program outcomes was a key consideration and concern for many organisations, with maintenance also a key indicator of the RE-AIM framework. To address this gap, additional data capture elements were built into the toolkit.

*Additional data capture*: A staff survey with associated real-time feedback for the champion on team level data (i.e., the ‘prize’ for completion) was added to the toolkit to assess sustainability. Like the pre- and post-program surveys, champions could monitor the number of responses (with tailored target response rates) and the data responses in real time. This sustainability staff survey became unlocked and visually highlighted (e.g., increased colour vibrancy from faded to full saturation) after the completion of the post-program survey. This design feature was intended to reduce the visible number of tasks and avoid incorrect data entry (i.e., from those not up to the maintenance assessment). A sustainability workplace audit was also included for the champion to complete. Both the staff survey and the audit included informational text about the benefits of evaluating long term changes at both the organisational team and individual level.

## 4. Discussion

This study described the application of a user-centred design process to optimise a peer-led behaviour-change program to be suitable for delivery and evaluation in the context of a national implementation trial. It provides insight into a thorough and iterative way of redeveloping an evidence-informed and usable online workplace health program delivered by workplace champions, using a “train-the-champion” approach. The champion dashboard, toolkit and associated collateral were redesigned to better meet the needs of workplace users, but also researchers and policy and practice partners who helped fund the program. Involving multiple stakeholders and end users, including experts from design and behavioural science, workplaces and industry was necessary to increase the likelihood that the program would be successfully implemented and evaluated during the next planned national implementation phase. The resulting minimal viable product or intervention adheres to UCD principles and best practice in behaviour change, and has been optimised for success according to all five RE-AIM dimensions.

Peer- or champion-led interventions offer a beneficial way of scaling evidence-based health behaviour programs delivered in the workplace [15,30]. If they are to reach their potential for wide scale impact, it is critical that researchers and the developers of such programs understand champion user needs and preferences, and respond to their priorities, concerns and challenges. Exploring ways to improve methods for involving workplace users (including champions, management and staff), as well other relevant stakeholders such as policy and practice partners, is important to ensuring programs are developed that are responsive to the contexts in which they are delivered [6], and thereby more likely to succeed. User centred design offers a methodology that helps achieve this, while also potentially making the best possible use of limited research funds by quickly focussing design and adaptations on data from end users. In the case of BeUpstanding, UCD aligned well with the participatory design principles [31] underpinning the program where champions are encouraged and supported to tailor the program to suit their team’s unique needs and existing culture.

Despite the ground breaking work of the Expert Recommendations for Implementing Change (ERIC) study [32] that highlighted best practice strategies in implementation research and practice, researchers have called for more attention to factors influencing uptake and more complete approaches to the promotion of implementation success [6]. UCD appears to be a useful approach to help researchers bridge limitations of existing frameworks (i.e., CFIR, ERIC) while encouraging interdisciplinary collaboration [6]. UCD can guide initial creation and subsequent adaptations of innovations by helping to highlight and address design problems related to the programs and their implementation strategies (such as low ease of use and high complexity). This in turn can impact uptake and effectiveness [7]. The usability of evidence-based health programs is a critical determinant of program outcomes including acceptability, feasibility, appropriateness and ultimately sustainability [33]. As such, it makes sense that users are placed at the centre of design and redesign efforts.

Use of the glossary of user-centred design strategies for implementation science by Dopp and colleagues (2019) [12] helped to highlight the multiple UCD strategies used during the redesign of the BeUpstanding toolkit and program. This was one of the first examples of a real-world application of UCD strategies according to the glossary, with 22 of the 30 strategies deployed. Some of the strategies outlined in the glossary such as ‘define target users and their needs’ were not relevant, as they had been completed prior to this phase of work. Other strategies such as ‘use of dialogic object-based techniques’ were too resource intensive to undertake. The glossary was not available when our redesign efforts were undertaken but it appears to be a useful tool to highlight the strategies undertaken, as well as to guide future redesign efforts to enhance implementation.

RE-AIM provided the other key framework that informed the redesign. To our knowledge, this is the first study that has integrated UCD methodologies to enhance RE-AIM indicators. Mapping design changes against this well-established framework ensured that changes made were going to meet the needs of the implementation trial. Our experience shows that RE-AIM can be used to organise redesign efforts, providing balanced focus particularly to issues of design that impact adoption and reach.

A critical element that informed the redesign was the iterative input gained from multiple sources (including end users, content experts, and policy/practice partners), using multiple data collection techniques. This occasionally conflicting feedback needed to be considered in the context of other considerations, including alignment with research aims and budgetary, technological and timeline constraints. Consequently, not all feedback was considered equally. Recently developed tools, such as the House of Quality for Behavioural Science [29], can be used to more systematically weight these perspectives from the different stakeholders to inform adaptations. This tool was published after our adaptation process, but the principles were in-line with our redesign decisions made. Deliberate consideration and documentation of how feedback should be weighted will help to ensure multiple user perspectives including ‘gatekeepers’ such as managers or decision makers in workplaces are considered [34], thereby advancing implementation and dissemination efforts [29].

While the formation of a multidisciplinary expert design team and involvement of stakeholders is a critical strength of UCD, the coordination of a large (10–15 person) team across diverse disciplines and holding distinct perspectives and needs was challenging. Relatedly, UCD involves a process that can sometimes be ambiguous, involving tolerance of differing opinions, pivots and repeated prototyping [35]. This process is somewhat in opposition to traditional research driven research methodologies [35]. UCD may not be feasible for all project timelines and resources particularly when stakeholders buy in and active engagement is difficult.

The redesign was used to optimise the BeUpstanding program and toolkit to be fit-for-purpose for the national implementation trial. This process highlighted the value in multidisciplinary expertise in helping to specify user and program needs, and make and test modifications. The trial will be completed in late 2022, and a similar UCD process is now been deployed to integrate the learnings from the trial to enable both scale out as well as scale up of the program. The core team are exploring the viability of the program through market research to ascertain funding models and potential commercial avenues to support the sustainability of the program in practice.

## 5. Conclusions

This paper described how UCD methodologies were applied to redesign an online, peer-led workplace health initiative to ensure it was fit-for-purpose for a national implementation trial. It highlights the multiple perspectives and data sources that were considered, and the personnel that were required, to ensure that the redesigned elements meet both the user needs and the research needs, according to the indicators of RE-AIM. These findings and reflections are intended to provide guidance to inform the design of other behaviour change research adaptations into practice.

## Figures and Tables

**Figure 1 ijerph-19-08096-f001:**
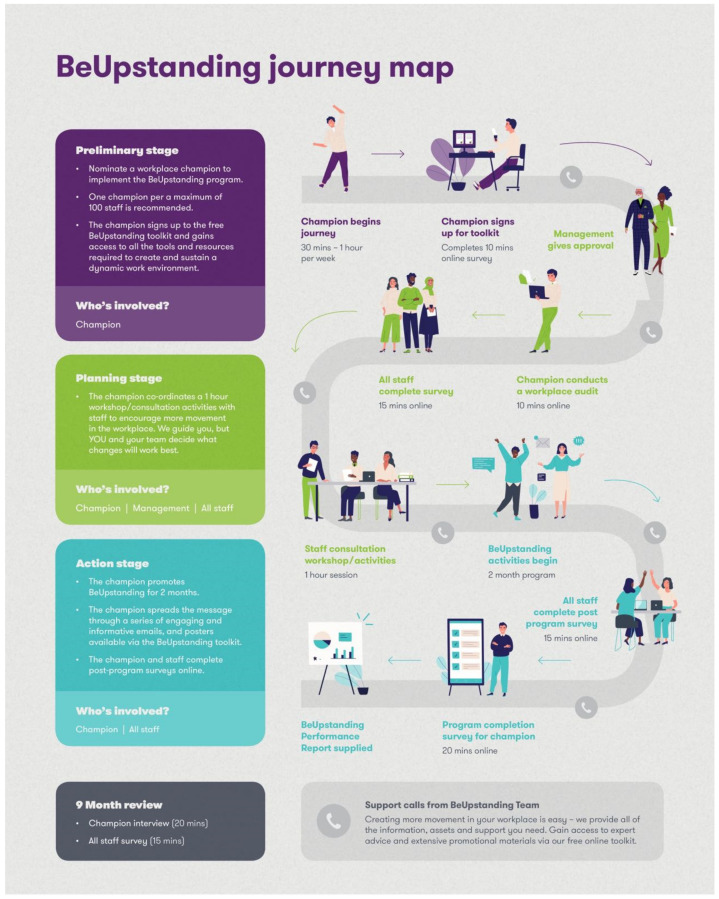
The BeUpstanding™ Journey Map© was one of the additional assets developed.

**Table 1 ijerph-19-08096-t001:** Primary stakeholders and end users involved in the redesign of the BeUpstanding program.

Stakeholder Group	Who They Were	How They Contributed
Core expert team	Design thinking and interaction design (SH) Business and product development (MF) Implementation science (AG) Content expertise (GNH) Software development	Led redesign and integration of changes within online toolkit
Policy and practice partners	SafeWork Australia (National Regulator) Comcare (National Work Health and Safety Authority) Office of Industrial Relations, Queensland VicHealth Healthier Happier Workplaces/Cancer Council WA	Identified priority needs of organisations and provided formal and informal feedback through stakeholder meetings and emails and telephone calls with core team
Workplace end users	Current, past and potential users of the toolkit (i.e., champions and overseers) Current, past and potential users of the program (i.e., staff)	Provided formal and informal feedback via online survey data, email and phone feedback, direct discovery interviews

**Table 2 ijerph-19-08096-t002:** Examples of the application of user-centred design strategies for the redesign and optimisation of the BeUpstanding program.

Term *	Example(s) of What Was Done
Apply process maps to system-level behaviour	Mapped all champion interactions that occur with the toolkit (e.g., journey map: when and how champion interacts with toolkit guide and resources), and staff interations with the program
Apply task analysis to user behaviour	Ideated and defined engagement strategies to be built into the toolkit for champions (e.g., incentives for completing tasks including customised champion certificate)
Collect qualitative survey data on users	Champion and staff surveys from soft launch included open text data collection around what worked well and barriers to implementation
Conduct co-creation sessions	Researchers and interaction designer mocked up prototypes of intervention elements (e.g., data reports) and sought feedback from users
Conduct design charette sessions with stakeholders	Members of the core team participated in intensive workshops to redesign program and toolkit elements
Conduct competitive user experience research	Business product development expert and researchers asked workplaces about other health and wellbeing programs they used or were aware of during discovery interviews
Conduct focus groups about user perspectives	Obtained management, champion and staff perspectives through discovery interviews
Conduct heuristic evaluation	Engaged design thinking and user-centred design expert to redesign the intervention toolkit and associated collateral (e.g., downloaded reports, information and tip sheets)
Conduct interpretation sessions with stakeholders	Discussion held at regular partner meetings concerning any conflicting perspectives of workplaces vs. partner/funders on desired look and feel and features of the toolkit and program
Conduct interviews about user perspectives	Obtained management, champion and staff perspectives on features
Conduct observational field visits	Observed workplaces through direct discovery interviews and field visits
Define target users and their needs	Core team identified and spoke directly with various stakeholders/users to redesign elements of the program and product based on problems they identified
Define work flows	Defined the process by which a champion takes up the program, enlists their team and delivers and evaluates it
Design in teams	Included interaction designer, software developer, business and product development expert and behaviour science experts in core team
Develop a user research plan	The research team planned this phase of work from the inception of the project with corresponding data collection methods, tools and personnel identified
Develop experience models	Profiles of workplaces were created (e.g., small with one team, large with multiple teams taking part in BeUpstanding)
Develop personas and scenarios	Profiles of the main users were created (i.e., researchers, overseers/management, champions, staff)
Engage in cycles of rapid prototyping	Mock-ups of the toolkit elements and collateral were created by the interaction designer and feedback sought
Engage in iterative development	Revised toolkit dashboard elements and collateral based on feedback and tested the generalisability of improvements/changes by asking different stakeholders to review
Examine automatically generated data	Objective user data (e.g., tasks completed by user champions) was collected automatically by the implementation platform
Prepare and present user research reports	Findings about the needs of each of the users were presented to the partners during regular stakeholder meetings and via emailed reports
Recruit potential users	Engaged users in different types of user research (via discovery interviews, workshops etc)to understand their needs, preferences and ideas for solutions

* Terms and associated definitions from Dopp et al., 2019 [12]. Glossary of terms and definitions for user-centered design strategies.

**Table 3 ijerph-19-08096-t003:** Priorities for redesign according to stakeholder/end user.

Stakeholder	Priorities for Redesign
Researchers	Capture information adequately to meet needs of implementation research trial Encourage uptake, engagement and fidelity with the program Program to be delivered and evaluated within budget constraints Maintain alignment with evidence-base and best practice in behaviour change
Policy and practice partners	Alignment with best practice in work health and wellbeing Encourage uptake in high priority ‘at need’ workplaces (i.e., small business, regional/rural, call centre and blue collar industries) Collection of data relevant to inform practice Provision of an evidence hub and a centralised resource they could refer workplaces to Provision of collateral and guidance to promote referral
Users of the toolkit (i.e., those delivering and evaluating the program)	More data and feedback that was easily digestible and compelling Change survey questions to be clearly relevant and as short as possible Increased guidance across the program related to structure and content of the program More engaging program materials for uptake, delivery and evaluation Streamlining the user experience and highlighting key/core components Adherence to a participatory approach whilst making the champion journey discrete and step based
Users of the program (i.e., staff taking part in BeUpstanding)	More collateral that was fun and engaging to maintain interest Wanted to feel visible support from management Concerns about data privacy and length of surveys Wanted more tips and tools

**Table 4 ijerph-19-08096-t004:** Modifications made to the BeUpstanding program and toolkit to ensure it was fit-for-purpose for national implementation and personnel support required to make the enhancements.

RE-AIM Dimension	Early Adopter Version Challenges	Improvements for Optimising the Program	Personnel Support Required to Make Enhancements
Reach	Lack of engaging materials to support champions to recruit and encourage staff	Revision of online and printed support materials to help champions invite and engage staff in the program (e.g., emails, posters).	Research team; interaction and graphic designer
	Inaccurate assessment of team numbers (a key denominator	Champions were given the ability to adjust and correct their initial data entry (provided in the champion profile survey) on their team numbers. Team numbers were visible in the survey portal and used to inform response rates.	Software developer
Effectiveness	Non-optimal staff survey response rate	Additional online content provided in the toolkit around the importance of evaluation. Desired response rates added to staff survey portal.	Research team; software developer
	Data feedback did not match expectations of champion/management end-user	Increased real-time feedback provided through staff survey portal. Incentive provided through bespoke reports for the workplace audit and following completion of the program completion survey.	Research team identified data points, graphic designer designed report, software developer integrated report.
Adoption	Limited business case for program Confusing champion journey	The free resources on the boarding page were refined and added to, including an animation of the program able to be shared with management.One page and two page infographics developed to capture time key actions and time commitment required from champions	Graphic designer, videographer, business consultants, research teamGraphic designer, research team and business developer
	Multi-stage onboarding process	Onboarding streamlined and simplified	Business developer, interaction designer, software developer
	Minimal recruitment channels with teams purposely approached and chosen by research staff	Development of a communications strategy (including recruitment goals, suggested target groups and recruitment avenues) and a communications package (including key content and graphics) for core partners to promote the program nationally to champions and worksites through existing networks and channels	Business consultants developed the comms and marketing strategy and package after consultation with each of the partners
Implementation	Program requirements and core steps not explicit	Development of champion journey infographics; dashboard redesigned to include more signposting and visual cues; collateral organised in weekly guide	Interaction and graphic designer; business consultants; research team
	Implementation data poorly captured	Addition of new survey for champions and new hard coded data entry with incentive (i.e., poster) to capture strategies and staff engagement in the workshop	Research team; software developer
Maintenance	No data captured	Staff survey portal (accessible by champions) expanded to include sustainability surveys. Sustainability audit (and report) added. Design features (e.g., lock and fade) to help avoid incorrectly times data completion incorporated into dashboard	Research team developed content; graphic designer developed report; software developer integrated into toolkit.

## Data Availability

Details on the early adopters can be found here: https://espace.library.uq.edu.au/view/UQ:8f73a27 (accessed on 11 April 2022). Other data (e.g., from discovery interviews; from meeting minutes) is identifiable and not able to be made publicly available, but is available on request and upon ethical approval from the corresponding author.

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
