# Peer review of "Applying a User Centred Design Approach to Optimise a Workplace Initiative for Wide-Scale Implementation"

_ijerph, 2022, doi:10.3390/ijerph19138096_

Round 1
Reviewer 1 Report
This paper described the effect of UCD methodologies on peer-led workplace health initiative, which could be fit-for-purpose for a national implementation trial. However, the results need more information, there were 113 workplaces participated in the survey, with only 7 of them providing staff response data and the staff survey rate of less than 43%, the author needs to analysis the reason for this situation. In addition, the suggestions were lack of the support of data, we suggest the author present the data of employee satisfaction, champion’s support data for different suggestions et al.
Author Response
Thank you for your comments.
We acknowledge that the data is presented differently than a typical results section. The intention was to highlight the process and the variations in the priorities by the different stakeholders, rather than the effectiveness findings (which have been reported elsewhere as now noted).
To clarify, there were multiple data collection points embedded within the online program, with 113 workplaces providing at least sign-up data. Staff survey data were obtained from 337 staff at pre-program and 167 staff post-program.
The seven workplaces discussed in this paper refer to those that were purposively sampled to provide feedback on their experience throughout the program. This feedback was additional to that which is collected from all workplaces through the online program. These seven workplaces, as well as an additional three workplaces who took part in deep dive interviews, along with the feedback collected through the toolkit, provided the insights presented regarding the poor completion of the staff surveys and helped inform the associated adaptations to the toolkit.
We have clarified further in the manuscript that in addition to the multiple methods and sources of feedback, we included multiple stakeholder perspectives.
“The process of gaining feedback was iterative, and included multiple stakeholders as outlined in Table 1.”.
Reviewer 2 Report
I thank you for the opportunity to comment this article. The subject is interesting.
Authors conclusion:
This paper described how UCD methodologies were applied to redesign an online, peer-led workplace health initiative to ensure it was fit-for-purpose for a national implementation trial. It highlights the multiple perspectives and data sources that were considered, and the personnel that were required, to ensure that the redesigned elements meet both the user needs and the research needs, according to the indicators of RE-AIM. These findings and reflections are intended to provide guidance to inform the design of other behaviour change research adaptations into practice.
Comments:
Main comment
The conclusion means according to my opinion that authors have created or improved methodology to design program. But this is not written like a methodological paper but rather authors state that they used UCD methodology for optimizing the toolkit. But I get confused because there are no references in the Methods. So, this dilemma needs to be solved. And if authors have used the certain methodology this needs to be referred too throughout this study.
Title
Title is too long and difficult
Abstract
Authors have tried to explain too many ideas in abstract and the result is that it is far away reader friendly. Kindly simplify and clarify.
Introduction
Introducing earlier Phases 1 to 3 is not appropriate. Authors can combine this earlier development in one paragraph. According to my understanding the different phases are not standardized and then presented the phases is not justified.
Results
I have difficulties to understand the purpose of Table 2. Justify it better or omit.
I think that Figure 1 needs to have copyright.
Author Response
Thank you for your comments. We have added in new references into the methods section and have added more citations of the seminal ‘IDEO design toolkit’ which details the methods of user centred design. It is important to note that we have not created a new methodology. Rather, this paper describes how we applied this methodology. Hopefully the addition of these references will clarify.
Title: Title is too long and difficult
We are happy to modify the title to “Applying a user centred design approach to optimise a workplace initiative for wide-scale implementation” as this retains the core purpose of the paper.
Abstract
Authors have tried to explain too many ideas in abstract and the result is that it is far away reader friendly. Kindly simplify and clarify.
We are happy to consider suggestions that the reviewer provides. We argue that the current abstract describes and defines the key terms and approaches used in the study. We are uncertain how we can retain this same level of information through modification.
Introduction
Introducing earlier Phases 1 to 3 is not appropriate. Authors can combine this earlier development in one paragraph. According to my understanding the different phases are not standardized and then presented the phases is not justified.
We respectfully maintain that the discussion of the different phases of the research to translation process provides important context to understanding the material presented in the current paper. It also highlights the importance of peer champions – which is the focus of the special issue. The different phases while not standardised, were deliberately planned as part of the research to translation process. They were also grounded in the well-established RE-AIM framework, as noted in the introduction.
Results
I have difficulties to understand the purpose of Table 2. Justify it better or omit.
The process detailed in the manuscript describes how the application of UCD methodology was used to inform the redesign of the program. We argue that it is important to summarise the UCD strategies as we have in Table 2 to provide an applied example for other researchers and practitioners who may be yet to apply UCD extensively during intervention design and/ or redesign.
We have modified our wording to make this clearer:
“A summary of the UCD strategies employed during the design phases are presented in Table 2. The UCD strategies were used to identify key intervention features and processes that required redesign, according to the terms and definitions by Dopp et al., 2019.12”
We have also added further detail:
“UCD strategies focussed on a broad array of characteristics including the individual (e.g., personas), the intervention (e.g., co-creation sessions and prototyping), as well as the organisation (e.g., observational field visits).”
I think that Figure 1 needs to have copyright.
Thank you, we have added in the Copyright statement.
Reviewer 3 Report
Dear Author
The article Applying a user centered design approach to optimizing an online, peer-led workplace initiative for wide-scale implementation based on
User centred design (UCD) methodologies this is an article useful for the community in general, however and because it is a research methodology which is opposed to the Traditional methodology, should be further explored with regard to:
Articulation of data collection techniques with objectives (should be improved)
Articulation of objectives with the results presented (should be improved)
Specify the criteria for inclusion in the study of the main parties involved in both organizations and participants.
Explain the criteria for the selection of experts
Explain how the scripts of direct interviews were made
Explain how field visits were recorded
Explain better how the data collection was done and what were the changes in the data collection platform.
Kind regards
Author Response
The article Applying a user centered design approach to optimizing an online, peer-led workplace initiative for wide-scale implementation based on
User centred design (UCD) methodologies this is an article useful for the community in general, however and because it is a research methodology which is opposed to the Traditional methodology, should be further explored with regard to:
Articulation of data collection techniques with objectives (should be improved)
Thank you for your comment. Due to the iterative nature of the design work, the data collection techniques overlap the objectives described in the introduction (which have now been modified based on your suggestion below). We have added detail in the Methods to more clearly highlight the data collection techniques that were used.
“The process of gaining feedback was iterative, and included multiple stakeholders as outlined in Table 1. Numerous methods10 were used to generate, collect, and synthesise ideas and feedback provided from the multiple stakeholders (workplace users; practice and policy partners; core expert team) across these three phases. These are described in detail below:”
Articulation of objectives with the results presented (should be improved)
Thank you for this comment we agree that out objectives could have been more clearly stated, in line with the results presented. We have modified the introduction as follows:
“Objectives are to: 1) provide examples of the application of UCD strategies for the redesign of the program and 2) highlight priorities for redesign according to stakeholders and 3) reflect on the process of optimization including the key elements required for success.”
We have also further highlighted the results in terms of our objectives in our results section:
“In line with our first objective, a summary of the UCD strategies employed during the design phases are presented in Table 2…”
“In line with our second objective, a summary of these priorities according to key stakeholder or user are described in Table 3…”
Specify the criteria for inclusion in the study of the main parties involved in both organizations and participants. Explain the criteria for the selection of experts
Thank you for your comments. We have included more detail about how the core team was formed and recruitment.
“A UCD methodology27 was applied to optimise the BeUpstanding Champion Toolkit for wide scale implementation and evaluation. The research team contracted experts in design thinking and interaction design (SH), and business and product development (MF), to form the core expert team which included an implementation scientist (AG). The core expert team led the iterative process of redesigning the toolkit. The primary stakeholders involved in the redesign were an expert interdisciplinary team that included foundational members involved in the research to practice translation process. These foundational stakeholders included a software developer; and state, national policy and practice partners. Workplace end users including champions, management and staff were also represented (Table 1). The toolkit’s online registration form provided details for ‘user’ organisations to be recruited via email, with ‘non user’ organisations recruited through partner channels via email.”
Explain how the scripts of direct interviews were made. Explain how field visits were recorded
Thank you for your comments. We have added more detail in the manuscript to answer both of these points better.
“The interviews were conducted in-person by members of the core team (and a project manager and honours level student). They followed a semi-structured interview script developed for the purposes of this work by the expert team (i.e., implementation scientist, content expert and product developer). Interviews were recorded using mobile application software and field visit notes were collected and compared across three visiting core team members”
Explain better how the data collection was done and what were the changes in the data collection platform.
We have added more detail to further elaborate, particularly for the online surveys used as detailed below and the direct discovery interviews as noted above.
“Online surveys: Open-ended responses in the online surveys (champions; staff) collected through the BeUpstanding toolkit and supporting implementation platform on what worked and didn’t work/could be improved were reviewed and synthesized”
We have also included additional information about changes to the data collection platform in the effectiveness section;
“Champions were also provided with the opportunity to view the survey prior to sending the survey link to staff.”
“Incentivising survey completion: Incentives were added to the toolkit to encourage completion of key self-report surveys including more detailed real-time data insights at baseline and follow-up survey times displayed for the champion (e.g. the common barriers staff reported to sitting less and moving more were displayed in a table). Tailored target response rates and feedback were also added (e.g., It’s great if at least 36/55 staff respond to this survey). “
Reviewer 4 Report
Dear authors,
Your paper presents a method of research and application of important resources in terms of performance and motivation of employees in terms of the health of the musculoskeletal system and cardio-respiratory system, with beneficial results on the health of the whole body.
I consider that the writing method is much too technical for this journal and it would be necessary to systematize and schematize the research design methodology, especially in the material and method section.
It would also be much easier for the reader to understand certain aspects if you described schematically the questions in the questionnaires used and the type of intervention applied.
Author Response
Thank you, however, we are unclear how to address this comment. There is no data presented on performance or motivation of employees, or health of the musculoskeletal system or cardiorespiratory system in this manuscript.
We have referenced our other published articles that do include details of the intervention, questionnaires and outcomes if this is what the reviewer is asking for.
Round 2
Reviewer 1 Report
Thank you for addressing my comments.
Good luck with your work.
Author Response
Thank you
Reviewer 2 Report
I thank you for the comments. I suggest that you shorten Table 2 and omit those issues which are not applicable.
Author Response
Thank you. We have removed the the methods that were not applicable from Table 2. We have also amended some of our descriptions in Table 2 to shorten the length.
Reviewer 3 Report
..
Author Response
Thank you